# Research on a Fault Diagnosis Method for Crankshafts Based on Improved Multi-Scale Permutation Entropy

**DOI:** 10.3390/s24030726

**Published:** 2024-01-23

**Authors:** Fengfeng Bie, Yu Shu, Fengxia Lyu, Xuedong Liu, Yi Lu, Qianqian Li, Hanyang Zhang, Xueping Ding

**Affiliations:** School of Mechanical Engineering and Rail Transit, Changzhou University, Changzhou 213164, China; bieff@cczu.edu.cn (F.B.); s21050858006@smail.cczu.edu.cn (Y.S.); lxd@cczu.edu.cn (X.L.); luyi@cczu.edu.cn (Y.L.); liqianqian@cczu.edu.cn (Q.L.); s21050858028@smail.cczu.edu.cn (H.Z.); s22050858042@smail.cczu.edu.cn (X.D.)

**Keywords:** crankshaft system, GA-MPE, PSO-SVM, fault diagnosis, pattern recognition

## Abstract

As the crucial part of a transmission assembly, the monitoring of the status of the crankshaft is essential for the normal working of a reciprocating machinery system. In consideration of the interaction between crankshaft system components, the fault vibration feature is typically non-stationary and nonlinear, and the single-scale feature extraction method cannot adequately assess the fault features, therefore a novel impact feature extraction method based on genetic algorithms to optimize multi-scale permutation entropy is proposed. Compared with other traditional feature extraction methods, the proposed method illustrates good robustness and high adaptability in the signal processing of crankshaft vibrations. Firstly, the improved complete ensemble empirical mode decomposition with adaptive noise (ICEEMDAN) method is developed on the signal to obtain several intrinsic mode function (IMF) components, and the IMF components with a large kurtosis are selected for array reorganization. Then, the parameters of multi-scale permutation entropy (MPE) are optimized based on genetic algorithm (GA), the multi-scale permutation entropy is calculated and the feature vector set is constructed. The feature vector set is input into the support vector machine (SVM) and optimized by a particle swarm optimization (PSO) model for training and final pattern recognition, where the Variational Mode Decomposition(VMD)-GA-MPE with a PSO-SVM recognition model and the ICEEMDAN-MPE with PSO-SVM recognition model without GA optimization are constructed for a comparison with the proposed method. The research result illustrates that the proposed method, which inputs the genetic algorithm optimized multi-scale permutation entropy extracted from the ICEEMDAN decomposition into the PSO-SVM, performs well in impact feature extraction and the pattern recognition of crankshaft vibrations.

## 1. Introduction

As the crucial component of reciprocating machinery, the crankshaft is responsible for delivering power and bearing impact loads [1]. Since the crankshaft is subjected to tangential and normal forces with periodic changes in magnitude and direction, even a minor shafting fault can result in a series of issues, including exacerbated vibrations, increased noise, and aggravated faults [2,3]. To increase the dependability of the mechanical equipment and guarantee the safe functioning of the equipment, it is essential to precisely assess the health state of the crankshaft system.

The vibration signal of a crankshaft system fault mainly consists of the periodic shock vibration components generated by faults and high-frequency inherent vibration components of the crankshaft system induced by shocks. Because the impact energy is relatively weak, the valuable low frequency information is frequently overwhelmed with other interference signals, thus it is required to investigate noise reduction and fault feature extraction of the crankshaft system vibration signal. The fault features of weak fault signals often fail to be captured by the traditional Fourier transform-based spectrum analysis method [4]. Although the valuable signal and noise can be effectively separated and extracted by decomposition and reconstruction methods, such as wavelet transform (WT) [5] and Wigner–Ville distribution (WVD) [6], there are many cross-interference items among the components, which reduce the time-frequency resolution and lead to local information distortion and detail loss [7]. In order to solve this problem, methods based on adaptive signal decomposition have been developed, such as the empirical modal decomposition (EMD) proposed by Huang et al. and the integrated empirical modal decomposition (EEMD) improved by Wu et al. on the basis of the EMD as a common decomposition method for nonlinear signals. Although it has been widely used in the field of signal processing, the EMD suffers from the problem of modal aliasing, and the EEMD is unable to completely neutralize the added Gaussian white noise, which occasionally renders an overall signal reconstruction error [8,9]. A method using a Variational Mode Decomposition (VMD) [10] algorithm was proposed in 2014, which overcomes the modal aliasing defects of EMD and minimizes the noise interference, but it comes with the drawback that the penalty parameter *α* and decomposition parameter *K* need to be determined manually based on experience, which depresses the efficiency of the signal processing. Therefore, due to the shortcomings exhibited by the above signal decomposition methods, based on the complete ensemble empirical mode decomposition with adaptive noise (CEEMDAN) [11] proposed by Torres et al., the improved complete ensemble empirical mode decomposition with adaptive noise (ICEEMDAN) proposed by Colominas et al. [12], which effectively suppresses the noise interference, greatly depresses the mode aliasing, and reduces the impact of spurious components on feature extraction.

In order to characterize the signal stochasticity and dynamical mutation characteristics, Yan et al. [13] introduced the permutation entropy into the rotating machinery vibration signal feature extraction, and related research shows that this feature can effectively characterize the working condition characteristics of rolling bearings in different states. Similar to the traditional single-scale nonlinear parameters, the alignment entropy can describe the irregularity of the time series on a single scale. However, since the vibration signals of shaft system failures are usually non-stationary and multicomponent signals, the single-scale feature extraction methods cannot ideally represent these signals. Aiming at the complexity and its changing law under multiple dimensions of the characterization time series, Aziz et al. [14] proposed the multi-scale permutation entropy (MPE) algorithm, with which the multi-dimensional information characteristics through further research could be extracted. The parameters required for the MPE in the computation process affect the results of the computation: the embedding dimension *m* determines the number of states contained in the phase-space reconstruction vector; the two-scale factor *s* is related to the entropy value obtained from the computation; and the magnitude of the value of *s* affects the degree of characterization of the signal eigenstates, which needs to be optimized [15]. Currently, researchers have proposed a variety of algorithms, including bat algorithms, ant colony algorithms, harmonic algorithms, particle swarm algorithms, fruit fly algorithms, imitation electromagnetism algorithms, genetic algorithms, and intelligent algorithms such as BP neural networks. Compared with other algorithms, genetic algorithms bring a faster global search capability and a wide adaptability, and have been applied in the optimization of various types of parameters [16]. In this paper, a genetic algorithm is introduced to solve the problem of MPE initial parameter setting.

Recently, many scholars have adopted various advanced intelligent classifiers to deal with those results obtained by the representative non-stationary signal processing methods, which are crucial for intelligent fault identification in practical engineering [17,18]. With the in-depth research of SVM in the field of fault diagnosis, intelligent fault diagnosis technology that combines signal processing methods with the SVM has been developed. Lyu proposed to combine VMD with SVM to extract the energy entropy as a feature vector input into SVM, which can accurately identify the bearing degradation state [19]. Zhao et al. proposed a method combining the local mean decomposition sample entropy with the SVM to solve the difficulty in extracting the fault features of a piston pump [20]. Xu et al. [21,22] proposed a novel fault diagnosis method based on data fusion, which could effectively diagnose equipment faults. Sharma used a binary discrete wavelet transform to decompose signals and extract their permutation entropy for the SVM classifier, which can effectively diagnose equipment faults [23]. When the SVM is applied in the fault diagnosis process, the kernel function, penalty factor *c*, and kernel function parameter *g* affect the accuracy of the diagnostic results. Since the particle swarm algorithm not only contains a fast convergence speed, it also does not require too many parameters to be set up, which makes it suitable for the parameter optimization of the SVM. This paper investigates an optimization based on a genetic algorithm of the multi-scale permutation entropy (GA-MPE) feature extraction, establishes the particle swarm optimization based on a support vector machine (PSO-SVM) model classifier, and inputs the collection of feature vectors of the crankshaft axle system in different states to the support vector machine for pattern recognition.

In order to effectively express the fault characteristics of the crankshaft shaft system, a novel feature extraction method based on a genetic algorithm optimized using multi-scale permutation entropy is proposed with comparison to the typical traditional feature extraction methods. The remainder of this paper is organized as follows: in Section 2, the theoretical basis is proposed, which introduces ICEEMDAN and the methods of GA-MPE and PSO-SVM models; Section 3 is the numerical simulation part, which describes the results of building the model, obtaining the simulation signal, and using the proposed method to improve the eigenvalues; Section 4 is the experimental verification part, where the experimental measurement of the signal and processing is used to verify the effectiveness of the method; and in Section 5, the major conclusions are presented.

## 2. Algorithm Principle

### 2.1. ICEEMDAN Algorithm

The ICEEMDAN is a further improvement of the CEEMDAN algorithm. The latter adds adaptive white noise at each stage of decomposition and obtains each modal component by calculating the residual. However, false noise components of the same scale will appear in the first few components of the decomposition. The ICEEMDAN improves this by adding white noise processed by EMD. This method greatly reduces the residual noise in modal components [24]. The specific decomposition process is as follows:(1)Add white noise $E_{k}\left(\omega^{\left(n\right)}\right)$ to the original signal x:(1)$$X_{i}^{\left(n\right)}=x+\alpha_{1}E_{1}\left(\omega^{\left(n\right)}\right),\left(n=1,2\ldots N\right)$$
(2)$$\alpha_{1}=\frac{\varepsilon_{1}\sigma \left(x\right)}{\sigma \left(E_{1}\left(\omega^{\left(n\right)}\right)\right)}$$
where x is the original signal, $\alpha_{1}$ is the expected signal-to-noise ratio of the signal for the first decomposition, and $\omega^{\left(n\right)}$ is the nth Gaussian white noise added.
(2)Calculate the first decomposition residual:(3)$$r_{1}=\langle X_{1}^{\left(n\right)}-E_{1}\left(X_{1}^{\left(n\right)}\right)\rangle$$
where $\langle •\rangle$ represents the sign of the average value.
(3)Calculate the first-order modal component $IMF_{1}$:(4)$$IMF_{1}=x-r_{1}$$(4)Calculate the kth-order residuals $r_{k}$:(5)$$r_{k}=\langle X_{i}^{\left(n\right)}-E_{i}\left(X_{i}^{\left(n\right)}\right)\rangle$$(5)Calculate the kth-order modal component $IMF_{k}$:(6)$$IMF_{k}=r_{i-1}-r_{i}$$(6)Let i=i+1 return (5) to calculate the next IMF.

### 2.2. Improved MPE

#### 2.2.1. Permutation Entropy

In 2002, Bandt and Pompe proposed the permutation entropy (PE) theory, which sorts complicated signals using symbols and determines the organization rule for each vector in phase space by reconstructing phase space. As a result, the complexity of the signal is determined by computing the probability of each symbol [25,26,27]. The calculation process is as follows:(1)Assuming $\left\{x_{i}\right\},i=1,2,\ldots N$ is a one-dimensional time series, phase-space reconstruction is performed on the one-dimensional time series:(7)$$Y=\left[\begin{array}{cccc} x_{1} & x_{1+t} & \cdots & x_{1+\left(m-1\right)t}\\ x_{2} & x_{2+t} & \cdots & x_{2+\left(m-1\right)t}\\ \vdots & \vdots & \vdots & \vdots \\ x_{k} & x_{k+t} & \cdots & x_{k+\left(m-1\right)t} \end{array}\right]$$
where m is the embedding dimension, t is the delay time, and $k=N-\left(m-1\right)t$.
(2)The components obtained by reconstructing each row of the matrix are sorted as follows:(8)$$Y_{i}=\left[x_{i},x_{i+t},\cdots ,x_{i+\left(m-1\right)t}\right],i=1,2,\cdots ,k$$(3)For an m-dimensional reconstructed phase space, each set $H\left(i\right)$ has a total of m! ordering possibilities. The number of occurrences of each sort are counted and their probabilities, $P_{1},P_{2},\cdots ,P_{k}$, are computed with:(9)$$\sum_{j=1}^{k}P_{j}=1$$(4)According to the defining equation of Shannon’s entropy, the permutation entropy is obtained as:(10)$$PE=-\sum_{j=1}^{k}P_{j}\ln\left(P_{j}\right)$$

#### 2.2.2. MPE

Information on other scales cannot be retrieved since the permutation entropy is limited in identifying the randomness of the one-dimensional time series. Aiming at this, MPE [28] was proposed to estimate the complexity of the time series at different scales. It is defined as the set of permutation entropies in the time series on different scales. Differing from the single scale permutation entropy, the MPE needs to coarsen the selected one-dimensional time series $\left\{x_{i}\right\},i=1,2,\ldots N$ with length N, and then the permutation entropy of each subsequence is achieved, which could be mainly described by the following steps [29]:(1)As shown in Figure 1, the time series $X=\left\{x_{i},i=1,2,\ldots ,N\right\}$ is divided into a window with a length of s.(2)The time series is coarsened as follows:(11)$$y_{j}^{\left(s\right)}=\frac{1}{s}\sum_{i=\left(j-1\right)s+1}^{js}x_{i},j=1,2,\ldots N$$
where s is the scale factor, s=1,2,…,N, and $y_{j}^{\left(1\right)}\left(j=1,2,\ldots ,N\right)\left(s=1\right)$ represents a coarse-grained time series with a length of N/s.
(3)An m-dimensional phase-space reconstruction is carried out for the coarse-granulated sequences $\left\{y_{j},j=1,2,\cdots ,m\right\}$:(12)$$Y_{l}^{\left(s\right)}=\left\{y_{l}^{\left(s\right)},y_{l+\tau }^{\left(s\right)}\cdots ,y_{l+\left(m-1\right)\tau }^{\left(s\right)}\right\}$$
where m denotes the embedded dimension, τ denotes the delay time, and l denotes the lth reconstructed component, $l=1,2,\cdots ,N-\left(m-1\right)\tau$.
(4)The reconstructed components are sorted in ascending order:(13)$$S\left(r\right)=\left(j_{1},j_{2},\cdots j_{m}\right)$$
where r=1,2,⋯k and k≤m!, which in turn calculates the probability of occurrence of any sequence of symbols, such that $P_{j}=\left\{j=1,2,\cdots ,k\right\}$, $\sum_{j=1}^{k}P_{j}=1$.
(5)The permutation entropy value of the coarse-grained lagging sequence is calculated using Equation (10), and then normalized through Equation (11) to obtain the MPE:(14)$$MPE=\frac{-\sum_{j=1}^{k}P_{j}\ln\left(P_{j}\right)}{\ln\left(m!\right)}$$

#### 2.2.3. Optimization of MPE Parameters by Genetic Algorithm

The parameters needed by the MPE in the computing process, such as the selection of the embedding dimension, m, and the scaling factor, s, will have an effect on the outcome [30]. In this paper, a genetic algorithm is employed in setting the initial parameters of the MPE. The global search ability of the genetic algorithm is used to find the optimal parameters of the MPE, including the embedding dimension, m, the scale factor s, the sequence length, N, and the delay factor, t. The processing steps are as follows:(1)Load the original signal, set the parameters to be optimized, and initialize the parameters of the multi-scale permutation entropy algorithm.(2)Take the square function of the multi-scale permutation entropy skewness as the fitness function, and calculate the skewness of the permutation entropy sequence $\left\{y_{j}^{\left(s\right)}\right\}$ under all scales of the time series $X=\left\{x_{i},i=1,2,\ldots ,N\right\}$ as follows:(15)$$Ske=\frac{E{\left[y_{j}^{\left(s\right)}-{\bar{y}}_{j}^{\left(s\right)}\right]}^{3}}{{\left[{\ddot{y}}_{j}^{\left(s\right)}\right]}^{3}}$$
where ${\bar{y}}_{j}^{\left(s\right)}$ is the mean of sequence $y_{j}^{\left(s\right)}$, ${\ddot{y}}_{j}^{\left(s\right)}$ is the standard deviation of sequence $y_{j}^{\left(s\right)}$, and $E\left(\cdot \right)$ is the expectation of the sequence.
(3)The objective function is:(16)$$F\left(X\right)=SKe^{2}$$(4)Carry out crossover, variation, selection and other operations.(5)Judge whether the termination conditions are met. If yes, output the results; otherwise, go to the previous step (4).

### 2.3. PSO-Based SVM Parameter Optimization

Vapnik et al. proposed the support vector machine algorithm based on statistical learning theory, which is a classification algorithm based on the concept of machine learning, statistical theory, and the principle of structural risk minimization [31]. The penalty factor, c, and the kernel function parameter, g, are important parameters that affect the accuracy of the SVM diagnosis results. Since a particle swarm algorithm not only has a fast convergence speed, it also contains less parameters to set, and so has good applicability for the parameter optimization of the SVM. Through cooperation and competition among individuals, the complex solution space is optimized to obtain the optimal solution of the problem [32,33,34].

The principle of the particle swarm optimization algorithm rests on the definition of a group of particles in the space. With each particle given a corresponding speed, position, and fitness of the initial value, the individual fitness value, $P_{best}$, and the extreme fitness value, $G_{best}$, of the initial population are calculated. After a period of movement, the local calculated $P_{best}$ and $G_{best}$ are applied to adjust their state and speed, seeking the optimal fitness. In the process of particle operation, namely the continuous iterative optimization, the particle updates its speed and position by comparing the fitness of the individual and global extrema. The update formula is:(17)$$v_{ij}^{k+1}=wv_{ij}^{k+1}+c_{1}r_{1}\left(P_{best,j}-x_{ij}^{k}\right)+c_{2}r_{2}\left(G_{best,j}-x_{ij}^{k}\right)$$
(18)$$x_{ij}^{k+1}=x_{ij}^{k}+v_{ij}^{k+1}$$
where $v_{ij}$ and $x_{ij}$ are the initial velocity and displacement of the ith particle, $P_{best,i}$, and $G_{best,i}$ is the optimal position of the ith particle and group passing by, w is the weight, $c_{1}$ and $c_{2}$ are the constant learning factors, $r_{1}$ and $r_{2}$ are random numbers within the interval $\left[0,1\right]$, and k is the number of iterations, 1≤i≤m, 1≤j≤d.

### 2.4. Troubleshooting Process

In this paper, a feature extraction method based on a genetic algorithm optimization of the multi-scale permutation entropy is proposed, in which an ICEEMDAN algorithm is developed to decompose the crankshaft vibration signal obtained in a numerical simulation into different IMFs, and then the IMF components with a larger kurtosis are selected for signal reconstruction. The entropy value is calculated based on a GA-optimized MPE for the reconstructed IMF components to verify the reliability of the feature extraction method. As for further experimental validation, the vibration signals of the crankshaft system in different fault states in the laboratory are measured. The entropy value extraction based on the ICEEMDAN decomposition of improved multi-scale permutation entropy model is completed and applied as the feature vector set. The feature vector set is finally input into the SVM model based on a PSO optimization for training and pattern recognition. The fault diagnosis research process is shown in Figure 2.

## 3. Numerical Simulation

The 3D model of the crankshaft system is created through SolidWorks according to the physical object, which is loaded into ANSYS through a good data interface for modal analysis and harmonic response analysis, and the dynamic properties of the crankshaft system are examined in order to verify the effectiveness of the method. The dynamic simulation of the crankshaft system is carried out by using Adams 2014 dynamic simulation software, through which the movement of the crankshaft system in the fault status is simulated and the desired acceleration signal is achieved. With the proposed method in this article, the crankshaft system fault characteristics are extracted effectively.

### 3.1. 3D Modeling of Shafting

Taking the crankshaft system of a 3H450 reciprocating piston pump as the analysis object, the dimensional parameters of this crankshaft system are shown in Table 1. Based on the important parameters of the crankshaft system above, the crankshaft, connecting rod, crosshead pin, and crosshead are modeled in the software SOLIDWORKS 2022, as shown in Figure 3, Figure 4, Figure 5 and Figure 6.

In order to simulate the failure mode of the crankshaft system, on the basis of a normal connecting rod, the wear failure of the connecting rod journal is regularized as a groove with a diameter of 712 mm, a width of 5 mm, and a depth of 2 mm, as shown in Figure 7. On the basis of a normal crankshaft, the crankshaft journal wear fault is regularized as an annular groove with a diameter of 708 mm, a width of 6 mm, and a depth of 2 mm, as shown in Figure 8. The normal crankshaft system and fault crankshaft system models assembled in SolidWorks were imported into ADAMS. The normal crankshaft system model is taken as an example, as shown in Figure 9.

### 3.2. Dynamic Simulation

The dynamic simulation of normal states, crankshaft faults, and connecting rod faults is carried out through the frame. Steel is selected as the model material. Considering the actual structure and running state, the model is completed in which the rotating pairs are applied to the crankshaft and the connecting rod individually, a fixed pair is added between the piston pin and the piston, a moving pair is added between the piston and the earth, and a contact force is applied between the connecting rod and the crankshaft journal. In this paper, the contact force between the crankshaft journal and the connecting rod is calculated based on Hertz contact theory [35]. The contact force parameters are shown in Table 2.

The decisive factor of the stiffness coefficient K is the material and structural shape of the impact object, which is calculated as:(19)$$K=\frac{4}{3}R^{\frac{1}{2}}E$$
(20)$$\frac{1}{R}=\frac{1}{R_{1}}+\frac{1}{R_{2}}$$
(21)$$\frac{1}{E}=\frac{1-\mu_{1}^{2}}{E_{1}}+\frac{1-\mu_{2}^{2}}{E_{2}}$$
where $R_{1}$ and $R_{2}$ are the radii of two contact bodies, respectively, $\mu_{1}$ and $\mu_{2}$ are the elastic modulus of two contact objects, and E is the equivalent elastic modulus of the object.

The rotating pair of the crankshaft should be driven for 3600d*time, with 1000 steps and a 1 s simulation time. The measured time-domain waveforms of the acceleration signals for the three axle system states (normal, connecting rod failure, and crankshaft failure) are shown in Figure 10.

The ICEEMDAN decomposition is performed on each of them to obtain a series of IMF components, and the decomposition diagrams are shown in Figure 11, where (a) is the normal state, (b) is the crankshaft failure, and (c) is the connecting rod failure. From the figure, it can be clearly observed that the modal components are significantly reduced and the signal characteristics can easily be extracted, which solves the problem of the appearance of spurious noise.

Taking the crankshaft fault as an example, in order to obtain the characteristic information containing only faults, the IMF components with obvious fault characteristics in the decomposition results are selected, and the kurtosis value of each component is calculated. The results are shown in Table 3.

The first four components with large kurtosis are selected for data reorganization. In order to obtain the characteristic information of the signals at different scales, the improved multi-scale permutation entropy is used to extract the characteristic values. A genetic algorithm is selected to improve the parameters of the MPE. The parameters include the embedding dimension, m, the scale factor, s, the sequence length N, and the delay factor, t. When the objective function is in the best fitness, its parameters are in the optimal state. The parameter diagram of the best fitness is shown in Figure 12. Here the best parameter values are obtained as m=5, s=7, N=147, and t=3. The multi-scale permutation entropy of the reconstructed and improved data of the three states is calculated for the feature vector set, as shown in Table 4.

The enhanced multi-scale permutation entropy of IMF_1_ under three states is displayed in Figure 13, which illustrates how drastically the GA-MPE under fault situations differs from the GA-MPE under normal conditions. The entropy value of the vibration signal under fault conditions is higher than that under normal conditions, indicating that it can be utilized as a characteristic index to characterize various crankshaft system states.

## 4. Experimental Verification

In this paper, the power end of the three-cylinder water injection pump, of the type BW 250, is employed as the research object, and the speed is 1200 r/min. The vibration signals are acquired by using devices such as handheld signal analyzers and accelerometers to identify the type of fault, and the sampling frequency is set to 5120 Hz. The block diagram of the vibration signal acquisition system is shown in Figure 14.

The time–frequency diagrams of the vibration signals measured by the accelerometers for the two states of the crankshaft (normal and wearing fault) are processed as shown in Figure 15. A total of 120 groups of data are extracted, with 2000 points sampled for each group of data. An amount of 60 groups of data for a normal crankshaft and 60 groups of data for a crankshaft with a fault are extracted, respectively. A total of 90 groups, or 45 groups from each state, were chosen at random to serve as training samples. A total of 30 groups comprise the remaining composition test samples.

Taking the crankshaft fault as the object, the signal is primarily decomposed by the ICEEMDAN into a series of IMF components, as shown in Figure 16. In order to obtain the characteristic information containing only faults, the IMF components with obvious fault characteristics in the decomposition results are selected, from which the kurtosis value of each component is calculated. The corresponding results are shown in Table 5.

The first six orders of IMF components with larger crags are selected for array reconstruction to improve the multi-scale permutation entropy parameter, and its optimal adaptation is shown in Figure 17, at which time its optimal parameter values are m=6, s=8, N=520, and t=2. The multi-scale permutation entropy of the reconstructed and improved data of the two states are calculated as the feature vector set. A group of data for the normal state and the fault state are taken as an example, and the calculation is shown in Table 6.

The crankshaft multi-scale permutation entropy under normal and fault conditions is calculated as a set of feature vectors, which are input into the PSO-SVM model for training and recognition, and the test takes 22.37 s. Figure 18 shows the fault pattern recognition based on the PSO-SVM. It can be seen that the recognition rate reaches 100%.

In order to validate the performance of the ICEEMDAN-GA-MPE and PSO-SVM fault identification model, the same vibration signals are applied as the input, the VMD-GA-MPE and PSO-SVM identification model, as well as the ICEEMDAN-MPE and PSO-SVM identification model without GA optimization, and the comparison of the computational results of each model are shown in Figure 19a,b. According to the figure, the fault recognition rate of the VMD-GA-MPE with PSO-SVM model is 93%, while the fault recognition rate of the ICEEMDAN-MPE with PSO-SVM model without GA optimization is merely 76%. The comparison demonstrates that it is preferable to input the genetic algorithm optimized multi-scale permutation entropy extracted from the ICEEMDAN decomposition into the PSO-SVM fault diagnosis method as the eigenvalue. In addition, the ICEEMDAN denoising effect has a significant impact on the model creation with an improved recognition rate.

## 5. Conclusions

In order to accurately capture the impacting features and identify the fault patterns of the crankshaft system in reciprocating machinery, a pattern recognition method based on a genetic algorithm optimization of multi-scale permutation entropy is proposed. The multi-scale permutation entropy model is enhanced using an ICEEMDAN to extract the entropy value, which is then input as a feature vector set into the SVM model based on a PSO optimization for training and pattern recognition. The outcomes demonstrate that the state characteristics of the crankshaft could be more accurately described by GA-MPE. When it is input into a PSO-SVM model as feature vector set, the accuracy rate reaches 100%. This work illustrates that the proposed method could solve the problem of the vibration features in the crankshaft system of reciprocating machinery being non-stationary and non-linear due to the interaction among components, and it is arduous to accurately express signal characteristics by the single-scale fault feature extraction method. This research provides a new effective way to diagnose faults in driven mechanisms (crankshaft systems) within reciprocating machinery.

In the work following this research, two crucial issues should be emphasized: (1), although the method proposed in this paper can obtain relatively ideal results, the processing efficiency and testing means need to be improved and enriched, and in the future novel and practical algorithms need to be involved in the model optimization; and (2), since the method could effectively realize the process of noise reduction and the nonlinear multi-scale feature extraction of the original vibration signals, it provides a greater application prospect in the fault diagnosis research field for machinery that includes similar driven mechanisms, which may inherently result in the valuable multi-scale features being overwhelmed by the background noise.

## Figures and Tables

**Figure 1 sensors-24-00726-f001:**
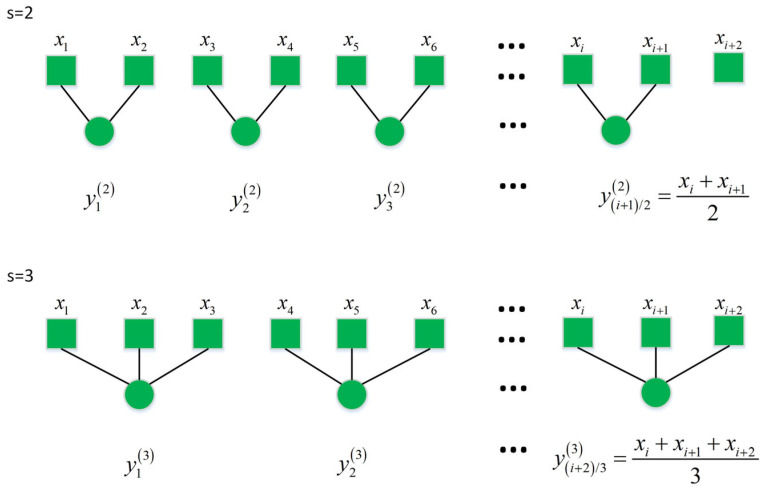
Schematic diagram of the MPE coarse granulation.

**Figure 2 sensors-24-00726-f002:**
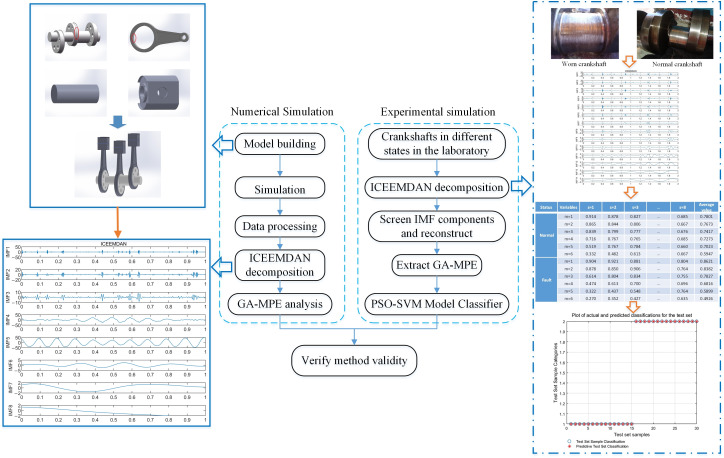
Troubleshooting flow chart.

**Figure 3 sensors-24-00726-f003:**
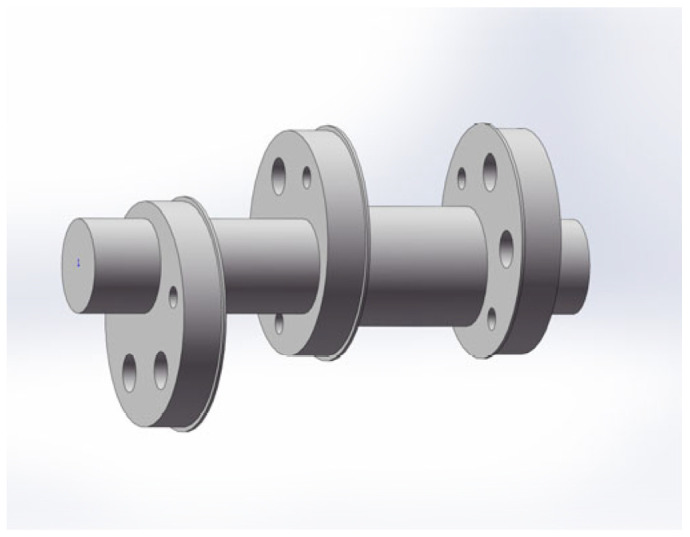
Crankshaft model.

**Figure 4 sensors-24-00726-f004:**
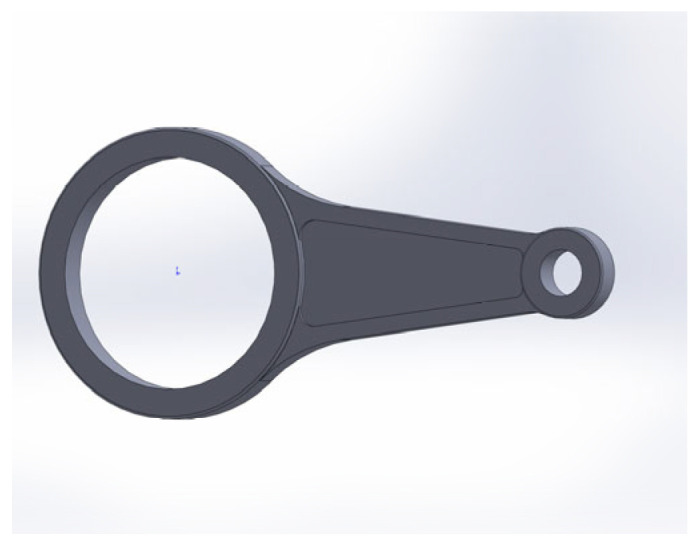
Connecting rod model.

**Figure 5 sensors-24-00726-f005:**
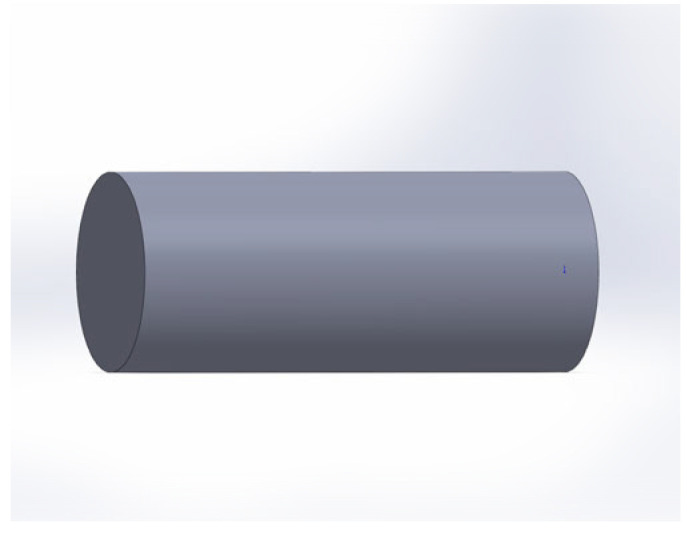
Crosshead pin model.

**Figure 6 sensors-24-00726-f006:**
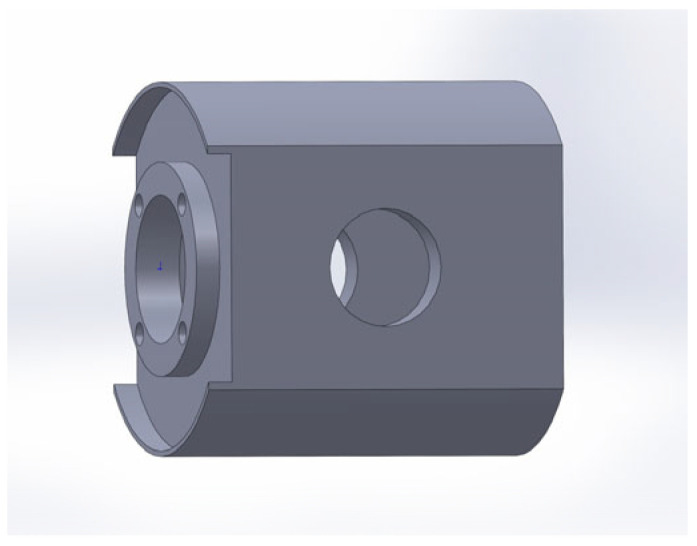
Crosshead model.

**Figure 7 sensors-24-00726-f007:**
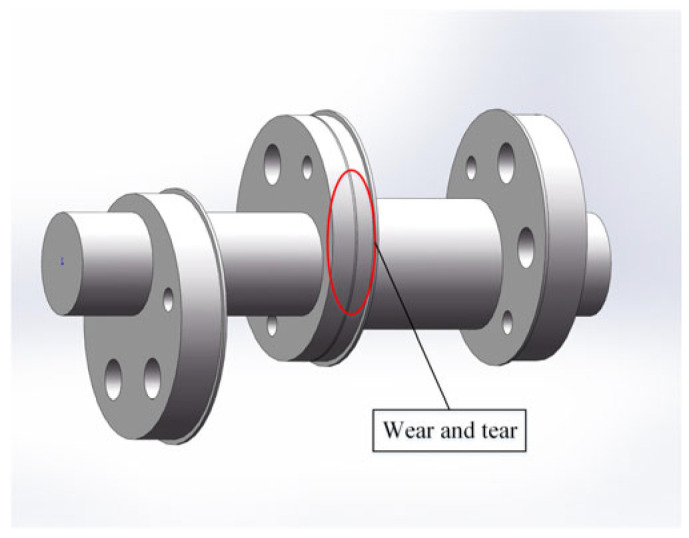
Crankshaft failure.

**Figure 8 sensors-24-00726-f008:**
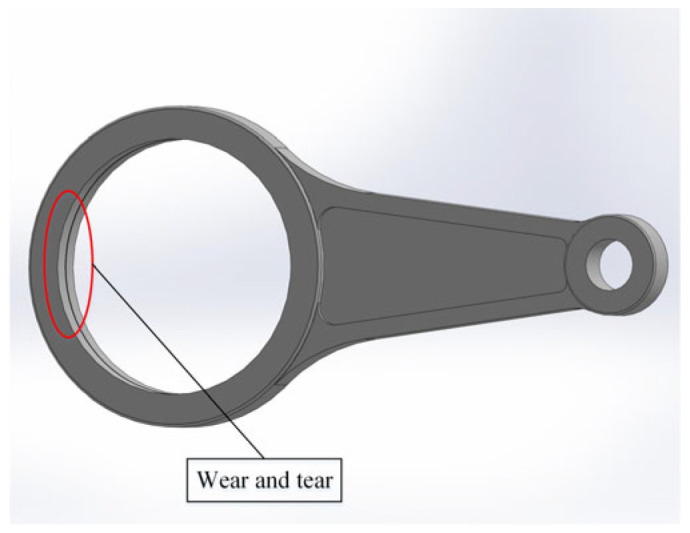
Connecting rod failure.

**Figure 9 sensors-24-00726-f009:**
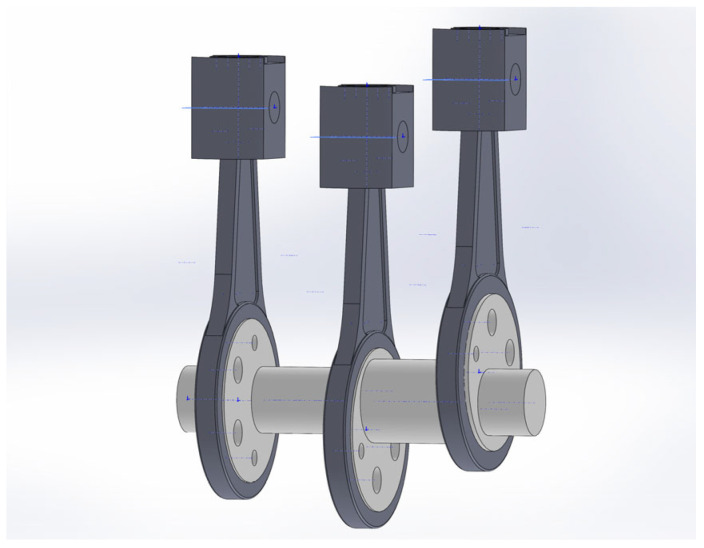
Model of a normal crankshaft system.

**Figure 10 sensors-24-00726-f010:**
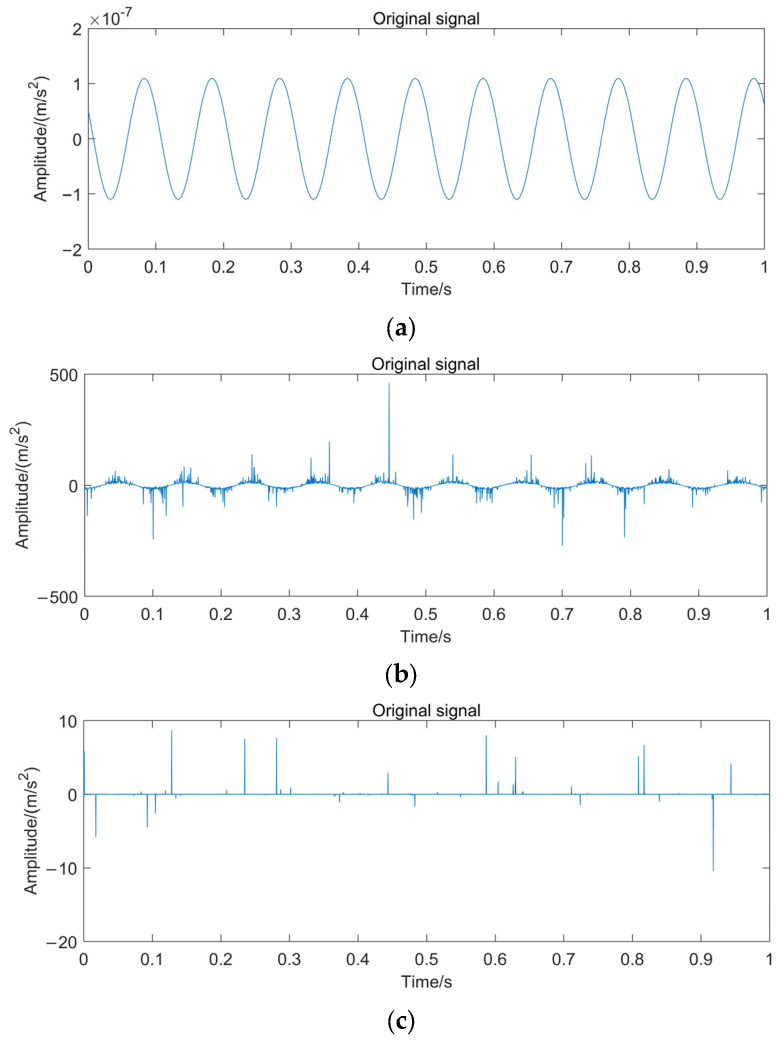
Time domain waveform of the simulation signal. (**a**) Normal shafting; (**b**) Crankshaft fault; and (**c**) Connecting rod failure.

**Figure 11 sensors-24-00726-f011:**
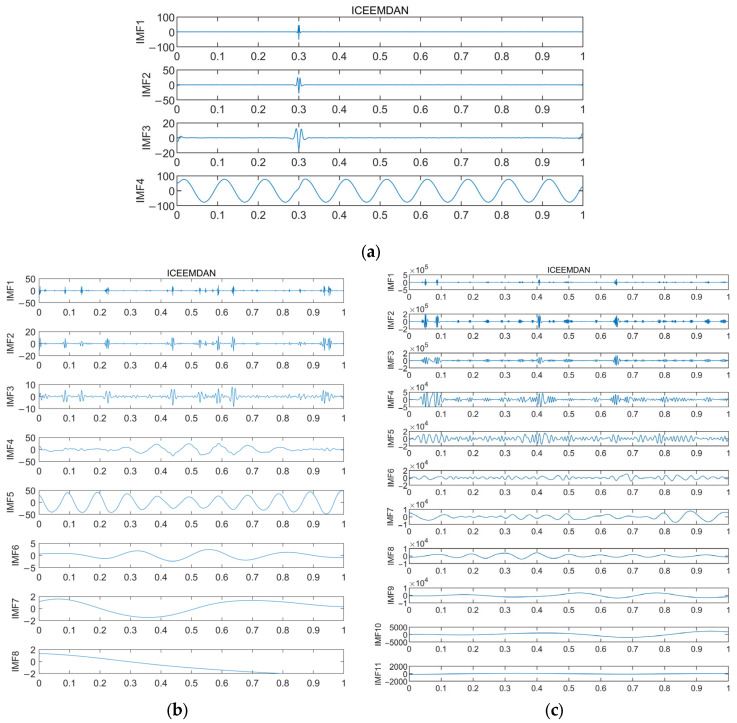
Exploded view of shaft system ICEEMDAN. (**a**) Normal shafting; (**b**) Crankshaft fault; and (**c**) Connecting rod failure.

**Figure 12 sensors-24-00726-f012:**
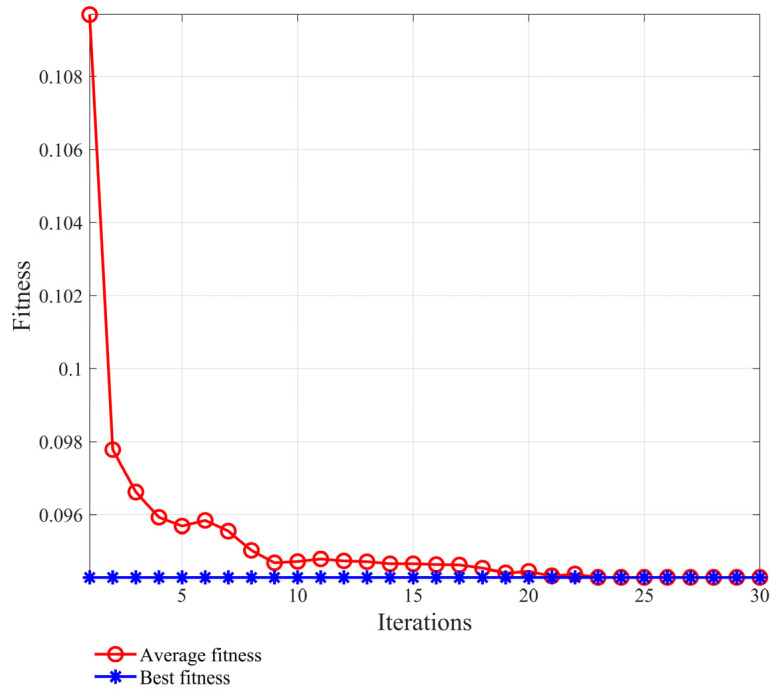
Crankshaft failure best fit trend graph (Analog signal).

**Figure 13 sensors-24-00726-f013:**
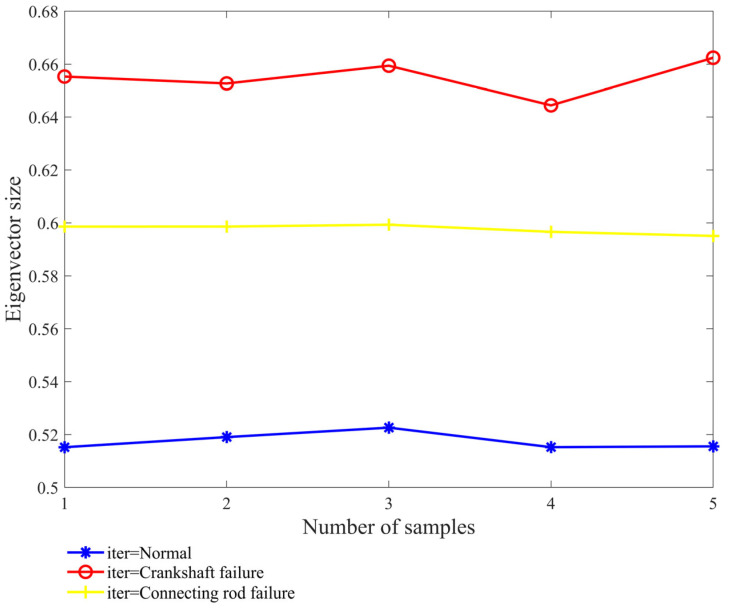
Multi-scale permutation entropy of IMF_1_ in three states.

**Figure 14 sensors-24-00726-f014:**
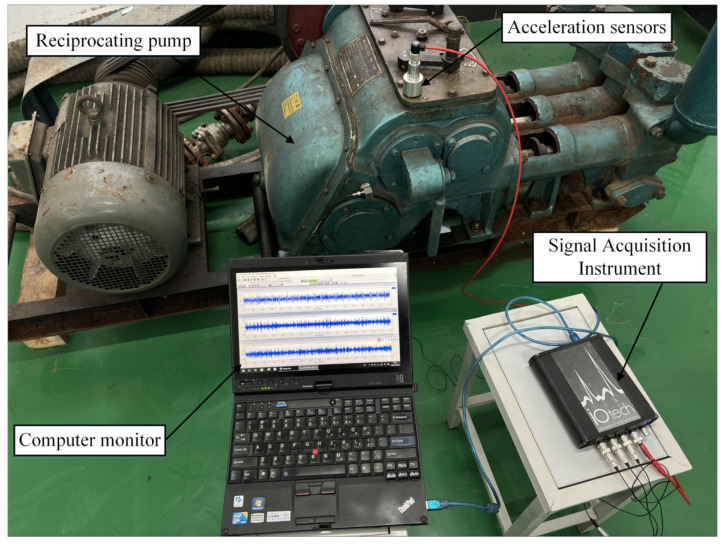
Testing platform of the reciprocating pump and the vibration acquisition system.

**Figure 15 sensors-24-00726-f015:**
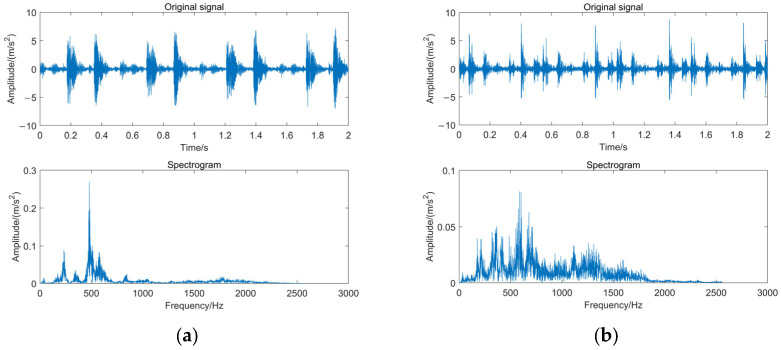
Time–frequency diagram of the vibration signal. (**a**) Normal shafting; and (**b**) Crankshaft with wearing fault.

**Figure 16 sensors-24-00726-f016:**
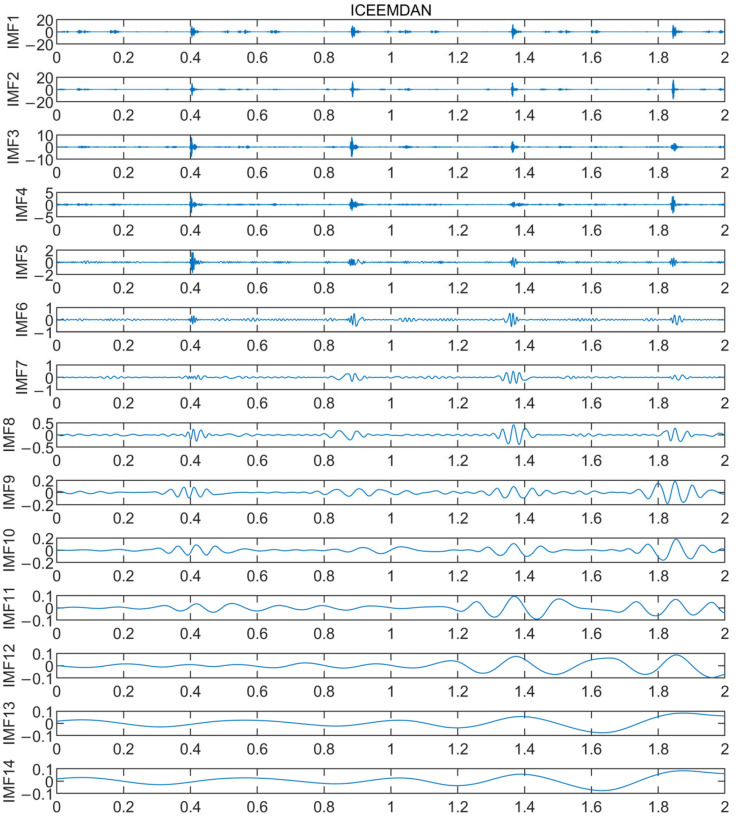
ICEEMDAN decomposition diagram of crankshaft faults.

**Figure 17 sensors-24-00726-f017:**
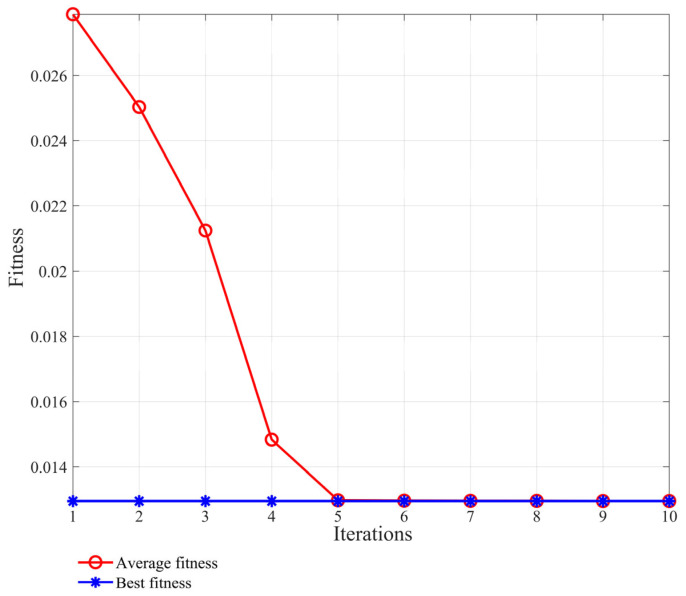
Crankshaft failure best fit trend graph (Experimental signal).

**Figure 18 sensors-24-00726-f018:**
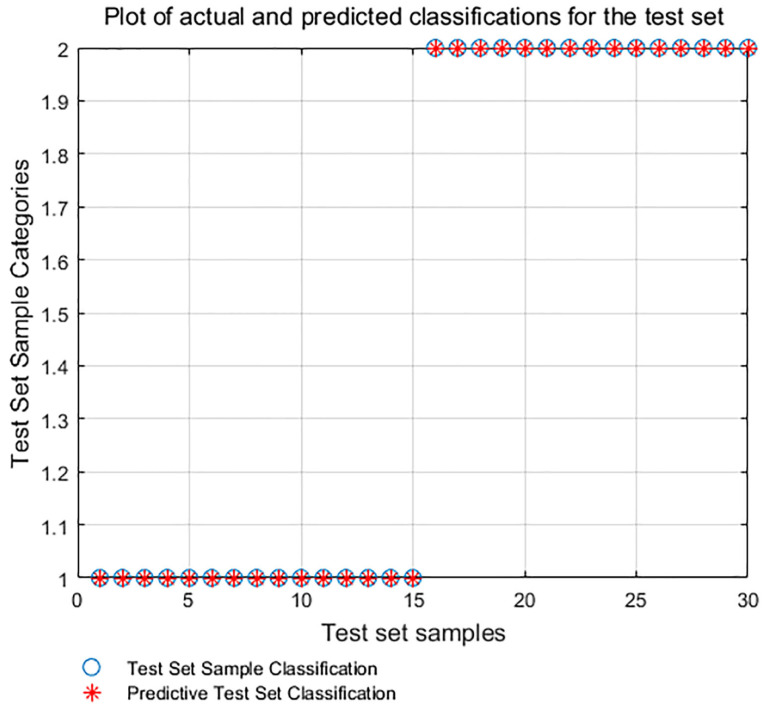
PSO-SVM-based identification diagram (ICEEMDAN-GA-MPE-PSO-SVM).

**Figure 19 sensors-24-00726-f019:**
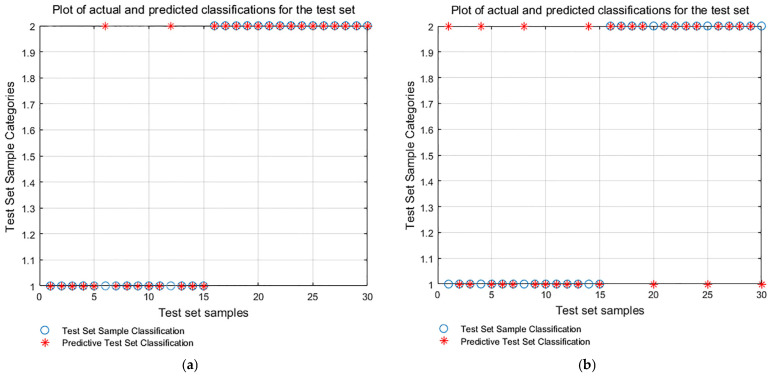
PSO-SVM identification results. (**a**) VMD-GA-MPE-PSO-SVM; and (**b**) ICEEMDAN-MPE-PSO-SVM.

**Table 1 sensors-24-00726-t001:** Main parameters of the crankshaft axis model.

Name	Size/mm	Name	Size/mm
Bearing journal diameter	300	Outer diameter of connecting rod big end	1100
Axial length of bearing journal	155	Axial length of connecting rod big end	200
Main journal diameter	325	Inner diameter of connecting rod big end	950
Axial length of main journal	153	Outer diameter of connecting rod small end	280
Inner diameter of eccentric	325	Axial length of connecting rod small end	175
Eccentric axial length	136	Inner diameter of connecting rod small end	143
External diameter of eccentric	740	Total length of connecting rod	2010
Diameter of outer arc end of crosshead	452.4	Internal length of crosshead arc	260
Outer diameter width of crosshead	295	Inner diameter width of crosshead	295
External axial length of crosshead	456	Internal axial length of crosshead	400
Crosshead pin diameter	143	Axial length of crosshead pin	343
Diameter of intermediate rod hole	180	Axial length of intermediate rod hole	30

**Table 2 sensors-24-00726-t002:** Contact force parameters.

Connecting Rod-Eccentric Wheel	Stiffness Coefficient	Damping Coefficient	Collision Parameters	Static Friction Factor	Dynamic Friction Factor
parameter	2.1 × 10^6^	2.1 × 10^4^	1.5	0.1	0.05

**Table 3 sensors-24-00726-t003:** Kurtosis value of the crankshaft fault for the IMF components.

Crankshaft fault	IMF_1_	IMF_2_	IMF_3_	IMF_4_
	22.959	16.227	8.733	3.316
	IMF_5_	IMF_6_	IMF_7_	IMF_8_
	1.683	2.249	2.128	1.760

**Table 4 sensors-24-00726-t004:** Eigenvectors.

State	IMF_1_	IMF_2_	IMF_3_	IMF_4_
Normal	0.5152	0.5196	0.4961	0.3628
	0.5191	0.5193	0.4974	0.3666
	0.5226	0.5221	0.4489	0.3628
	0.5152	0.5165	0.4833	0.3628
	0.5155	0.5180	0.5048	0.3628
Crankshaft wear	0.6553	0.6521	0.6379	0.5061
	0.6527	0.6359	0.6295	0.4937
	0.6594	0.6533	0.6410	0.4250
	0.6444	0.6511	0.6284	0.4362
	0.6624	0.6364	0.6154	0.4013
Connecting rod wear	0.5986	0.5944	0.5647	0.5196
	0.5986	0.5954	0.5665	0.5312
	0.5993	0.5941	0.5636	0.5244
	0.5966	0.5947	0.5756	0.5343
	0.5951	0.5950	0.5613	0.5256

**Table 5 sensors-24-00726-t005:** Kurtosis values for two shaft systems for the IMF components.

Crankshaft fault	IMF_1_	IMF_2_	IMF_3_	IMF_4_	IMF_5_	IMF_6_	IMF_7_
	52.062	132.520	77.403	53.159	42.920	15.252	12.807
	IMF_8_	IMF_9_	IMF_10_	IMF_11_	IMF_12_	IMF_13_	IMF_14_
	11.172	7.186	6.388	4.256	3.465	2.860	1.618

**Table 6 sensors-24-00726-t006:** Eigenvector calculation table.

State	Variable	s = 1	s = 2	s = 3	s = 4	s = 5	s = 6	s = 7	s = 8	Mean Value
Normal	m = 1	0.914	0.878	0.827	0.768	0.740	0.734	0.696	0.685	0.7801
	m = 2	0.865	0.844	0.806	0.776	0.760	0.715	0.707	0.667	0.7673
	m = 3	0.839	0.799	0.777	0.741	0.742	0.700	0.659	0.676	0.7417
	m = 4	0.716	0.767	0.765	0.760	0.731	0.714	0.681	0.685	0.7273
	m = 5	0.519	0.767	0.784	0.756	0.726	0.714	0.692	0.660	0.7023
	m = 6	0.332	0.482	0.613	0.639	0.679	0.668	0.678	0.667	0.5947
Fault	m = 1	0.904	0.921	0.881	0.907	0.859	0.827	0.793	0.804	0.8621
	m = 2	0.878	0.850	0.906	0.836	0.803	0.840	0.828	0.764	0.8382
	m = 3	0.614	0.804	0.834	0.795	0.833	0.809	0.818	0.755	0.7827
	m = 4	0.474	0.613	0.700	0.762	0.809	0.742	0.657	0.696	0.6816
	m = 5	0.322	0.437	0.548	0.584	0.626	0.722	0.716	0.764	0.5899
	m = 6	0.270	0.352	0.427	0.473	0.536	0.593	0.655	0.635	0.4926

## Data Availability

All the data in this article are obtained by the author’s field measurement. If necessary, please contact the author by email.

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
