# Peer review of "Research on a Fault Diagnosis Method for Crankshafts Based on Improved Multi-Scale Permutation Entropy"

_sensors, 2024, doi:10.3390/s24030726_

Round 1

Reviewer 1 Report

Comments and Suggestions for Authors

Reviewer 2 Report

Comments and Suggestions for Authors

1) The motivation of this study should be improved. In abstract, the author declare that single-scale feature extraction method cannot adequately express the fault features, so they put forward this approach. However, as far as i am conerned, many multiscale models have been developed for fault detection.

2) In introduction section, the motivations and advantages of this approach are recommended to be listed so that it is more comprehensive for readers.

3) Why did the authors apply genetic algorithm to optimize the MPE parameters. How about other optimization approaches?

4) What about the computational efficiency of the developed approach?

5) More relaetd approaches such as  

Online Knowledge Distillation Based Multiscale Threshold Denoising Networks for Fault Diagnosis of Transmission Systems,

CFCNN: A novel convolutional fusion framework for collaborative fault identification of rotating machinery.

6) The mechanism of multi-scale permutation entropy should be described in more detail

7) Future work should be prospected in the conclusion section.

Reviewer 3 Report

Comments and Suggestions for Authors

This paper proposed a fault diagnosis method for crankshafts. This method combines the multi-scale permutation entropy optimized by genetic algorithm with PSO-SVM.

      This article cannot be accepted for publication because this article lacks innovation. The methods used in this paper, such as improved complete ensemble empirical mode decomposition with adaptive noise (ICEEMDAN), Multiscale permutation entropy (MPE) algorithm, genetic algorithm (GA), and support vector machine (SVM) optimized by particle swarm optimization (PSO) model, are all proposed by others. These methods have also long been applied to the field of fault diagnosis. All that is done in the paper is to optimize the parameters of the MPE algorithm with genetic algorithm (GA). The novelty of this paper is not enough.

      The article cannot be accepted for publication.

Comments on the Quality of English Language

NA

Round 2

Reviewer 1 Report

Comments and Suggestions for Authors

The study uses a genetic algorithm (GA) to optimize the parameters of multi-scale permutation entropy (MPE), and particle swarm optimization (PSO) to optimize SVM. The need for the methodology proposed in the paper is not sufficient and its advantages cannot be adequately demonstrated. However, the paper applies the proposed method to solve the engineering problem of condition detection of reciprocating pumps, and the results verify the advancement of the proposed method, based on which the paper is recommended for acceptance and publication

Reviewer 2 Report

Comments and Suggestions for Authors

This paper is ready for publication.

Reviewer 3 Report

Comments and Suggestions for Authors

The expression "the fault vibration signals are typically non-stationary and nonlinear" is wrong. A signal can not be nonlinear. 

Please consider Please consider enhancing the lit review by including more intelligent fault diagnosis methods, such as doi: 10.1109/TIM.2023.3259048, 10.1016/j.ymssp.2022.108907.

Comments on the Quality of English Language

NA
